# Magnesium as an Important Factor in the Pathogenesis and Treatment of Migraine—From Theory to Practice

**DOI:** 10.3390/nu14051089

**Published:** 2022-03-05

**Authors:** Izabela Domitrz, Joanna Cegielska

**Affiliations:** Department of Neurology, Faculty of Medical Sciences, Medical University of Warsaw, 01-809 Warsaw, Poland; joanna.cegielska@wum.edu.pl

**Keywords:** migraine, neural hyperactivity, CSD, NMDA receptor, magnesium deficiency

## Abstract

So far, no coherent and convincing theory has been developed to fully explain the pathogenesis of migraine, although many researchers and experts emphasize its association with spreading cortical depression, oxidative stress, vascular changes, nervous excitement, neurotransmitter release, and electrolyte disturbances. The contribution of magnesium deficiency to the induction of cortical depression or abnormal glutamatergic neurotransmission is a likely mechanism of the magnesium–migraine relationship. Hence, there is interest in various methods of assessing magnesium ion deficiency and attempts to study the relationship of its intra- and extracellular levels with the induction of migraine attacks. At the same time, many clinicians believe that magnesium supplementation in the right dose and form can be a treatment to prevent migraine attacks, especially in those patients who have identified contraindications to standard medications or their different preferences. However, there are no reliable publications confirming the role of magnesium deficiency in the diet as a factor causing migraine attacks. It also seems interesting to deepen the research on the administration of high doses of magnesium intravenously during migraine attacks. The aim of the study was to discuss the probable mechanisms of correlation of magnesium deficiency with migraine, as well as to present the current clinical proposals for the use of various magnesium preparations in complementary or substitute pharmacotherapy of migraine. The summary of the results of research and clinical observations to date gives hope of finding a trigger for migraine attacks (especially migraine with aura), which may turn out to be easy to diagnose and eliminate with pharmacological and dietary supplementation.

## 1. Introduction

Migraine is a neurological disease characterized by severe attacks of headache with hypersensitivity to light, sounds, and most often with nausea and vomiting [1]. This disease reduces quality of life and weakens production capacity [2,3]. It is also worth mentioning that migraine is the second leading cause of disability worldwide, but there is a paucity of strong data on migraine-related illness affecting work performance [4].

In addition, it is widely known that migraine prevents young people from engaging in their professional, social, and personal activities to their best ability [5]. That is why correct, effective, and non-burdensome treatment of migraine is so important. Proper treatment and prevention can reduce the burden on healthcare and improve the quality of life of patients.

Some medications recommended in migraine prophylaxis by the International, European, and local Headache Societies and federations involve magnesium in dealing with migraine. Additionally, many working and active patients do not want to take medications and prefer to use supplements, including magnesium. Magnesium is recommended by the above-mentioned organizations, but at a lower level—C or U [6,7,8,9]. 

A migraine attack can be initiated by many triggers, such as eating or not eating a specific food, exercise, weather changes, and others [10]. Many of them have been reported as triggers of migraine in previous scientific studies [11,12]. A study by Chądzyński et al. has also recently confirmed that we can identify these triggers as related to the appearance of headaches in people with migraine. One such factor may be hypomagnesemia. Fila et al. [13] concluded that several nutrients (including magnesium) can be considered as dietary supplements to prevent and/or alleviate migraine attacks by improving mitochondrial function and energy production in the brain and reducing oxidative stress, which can be one of the processes underlying the pathogenesis of migraine. 

Due to the above argumentation and other known migraine hypotheses, the aim of this review was to evaluate the scientific evidence of a significant role of hypomagnesemia in the pathogenesis of migraine and to seek/find a rationale for the use of magnesium in the treatment of migraine. Our study, as a narrative review, was based on the published articles mainly in recent years. We considered the role of magnesium in the pathogenesis of migraine and therefore its usefulness in the migraine treatment. For this purpose, we took into account the articles that dealt with the above problem and, at the same time, based on the current studies of on the pathogenetic role of magnesium.

## 2. The Pathogenesis of Migraine and the Magnesium Role

Despite the knowledge accumulated so far, the pathogenesis of migraine has not been fully elucidated and understood yet. During the last seventy years, many theories have been developed about migraine pathogenesis. Some vascular neuronal theories with neurotransmitters changes and disturbances in the functions of ion channels and numerous types of receptors (NMDA, AMPA, mGluR, cannabinoid, vanilloid, and PAR), as well as the triggering mechanisms and the course of the process of neurogenic inflammation are important and constantly analyzed. Many clinical and experimental observations have identified abnormalities in vasoactive neurotransmitters and peptides releasing in migraine but coherent concept has not been explained the triggering mechanism of migraine in detail [14,15]. Some studies with phosphorus magnetic resonance spectroscopy show altered energy metabolism in the brain. This abnormality may be related to genetic, neuronal, and vascular factors. It is known that migraine is associated with a mutation in genes coding metabolic enzymes both mitochondrial and nuclear DNA [16,17,18,19].

Cortical spreading depression (CSD) and, in some cases, cerebral flow decreasing during a migraine attack are only partly explaining the migraine symptoms background [20,21,22,23,24,25,26]. The self-propagating wave of transient CSD can underlie the pathogenesis of migraine, especially its aura. However, the mechanism of its triggering in the structurally and nutritionally unaltered cerebral cortex of migraine patients remains unclear. Some internal and/or external factors such as light, strong odors, noise, substances contained in certain foods or electrolyte disturbances (e.g., hypomagnesemia), are known as trigger factors for migraine attack. As early as 1990s, Olesen et al. confirmed the role of potassium in CSD, in activation of the cerebral vessels and irritation of the perivascular nerves and also in neurotransmitters released [27]. It has been hypothesized that the transient destabilization of the neuronal excitatory–-inhibitory balance allows external or internal factors to overactivate the cerebral cortex, which, in turn, creates the conditions for initiating a wave of cortical depression. 

According to the results of Vinogradova research [27] and, in earlier years, Van Harreveld’s study [28], the triggering factor may be long-term depolarization of neurons associated with the excessive calcium-dependent release of glutamate or an increase extracellular potassium concentration. It cannot be ruled out that both factors occur simultaneously. 

Martens-Mantai et al. [29] proved that the effects of various compounds, e.g., the injection of a KCl solution, and tetanus electrical stimulation of the entorhinal cortex affect propagation of cortical depression (CSD) from the neocortex to the hippocampus. Also, the use of the *N*-methyl-d-aspartate (NMDA) receptor blocker and GABA A receptor blocker in the entorhinal cortex area decreased CSD transmission to the hippocampus. 

On the other hand, the structure of the NMDA receptor involved in glutamate release, may play an important role in initiating cortical depression. The proper magnesium and calcium metabolism is important for the functioning of this receptor, and disturbances of ions concentration can affect the induction of cortical depression. One of the main functions of magnesium in the nervous system is its interaction with the NMDA receptor. The magnesium ion blocks the calcium channel in the NMDA receptor and protects the cell against the uncontrolled influx of calcium ions.

Therefore, hypomagnesemia can enhance glutamatergic neurotransmission and promote excitotoxicity and consequently lead to oxidative stress. 

It was found that lowering the level of magnesium in the platelets of patients with migraine is associated with an increase level of cyclic AMP, but is not related to the concentration of cyclic GMP. The levels of those substances, in turn, are probably related to the release of neurotransmitters responsible for vasomotor disorders during the initial phase of migraine and the appearance of the cortical depression (CSD) [30,31,32]. Hypomagnesemia may also reduce the gating of nociceptive sensations in the spinal centers, which probably is directly related to the headache in the course of migraine attack. It is believed that neurological disorders such as migraine, chronic pain, and epilepsy are related precisely to abnormal glutamatergic neurotransmission [30,33,34,35,36].

The potential role of magnesium in the pathogenesis of migraine is schematically shown in Figure 1.

## 3. The Role of Magnesium in Migraine Pathogenesis

Changes observed in migraine (especially migraine with aura), such as disturbances in regional cerebral flow and cortical spreading depression, and oxidative stress with the activity of the brainstem [40,41,42] might be connected with electrolyte and magnesium disturbances, which may play a key role in the pathogenesis of migraines. In recent years, there has been growing interest in the possible role of magnesium deficiency in initiation of migraine attacks. The effects of its acute and chronic administration (intravenous or oral) are assessed in terms of the relief of migraine headaches and symptoms of hypersensitization [1,33]. The role of magnesium is the best-known in menstrual migraines [43]. 

Magnesium is involved in various biological processes. It acts as a cofactor for synthase appropriate to produce ATP, regulates neuronal excitability and also plays an important role in maintaining the tone of blood vessels [44]. Magnesium is involved in over 600 enzymatic reactions in human. Its function is mainly connected with production and storage of high-energy phosphates. [45]. Magnesium ions block N-methyl-d-aspartate (NMDA) receptors and in this mechanism inhibit neuronal calcium flow, so it may play a role in the initiation and maintenance of central sensitization after nociceptive stimulation [13]. CSD persistent during migraine attacks is connected with NMDA receptors activation, so these receptors might be a target to migraine treatment. The important pathway may contribute modulation intracellular proinflammatory process. Also, magnesium may keep the balance of neurotransmitter release, platelet activity, and vasoconstriction [46]. CGRP level also appears to be related to magnesium [47,48,49,50]. 

In short, magnesium, in the form of ions present inside and extracellularly, is essential for brain and other organs energy homeostasis. Its deficiency is an evidenced risk/trigger factor for migraine attack.

## 4. The Rationale for Magnesium Treatment for Migraine

There are about 25–35 g in the human body magnesium, of which about 53% is stored in the bones, 46% in muscles and soft tissues, and only 1% in the blood [51]. The data published so far on the concentration of magnesium in the blood in people with and without migraine is not uniform. Some of migraine patients have lower serum levels of magnesium as compared with the non-migraineurs control. 

According to some researchers, migraine patients with severe headaches have lower magnesium levels than their counterparts with mild to moderate headaches [52,53,54,55,56,57,58]. On the other hand, some studies reported normal magnesium levels in blood serum in migraineurs. For instance, this is true of the study of Cegielska et al.; although in all subjects enrolled in the study the magnesium level was within normal range, the electrophysiological tests showed pathology applicable to hypomagnesemia [59]. The authors presented the potential correlation of the frequency and severity of migraine with disturbances of neuromuscular excitability dependent on the level of intracellular magnesium. It is known that intracellular magnesium levels do not correlate indirectly with general serum magnesium levels.

Concerning the usefulness of the use of magnesium in the treatment of migraine, conclusions from the Kaur et al. study can be quoted [16]. Authors concluded that antioxidants appear to be helpful in migraine prevention. They showed that magnesium and other supplementation factors might reduce the number of days of migraine attacks. A few other studies also showed a decrease in the intensity and duration of migraine headaches. However, more experiments and observation, on larger material is needed to definitively conclude the effects of these substances on the prevention and effective treatment of migraine attacks. Based on the above research, and also two systematic reviews with meta-analysis, four randomized placebo-controlled trials double-blinded, three open-label pilot studies, one observational study, one cohort study, and review articles, it can be concluded that the role of magnesium in the pharmacotherapy of migraine is undoubted, although its dosage, the type of effective substances and the duration of their use remain unclear. It can certainly be recommended. However, so far, we have not received evidence to recommend such treatment in a class higher than C. Magnesium is recommended by the international and local experts at a lower level—C or U, as mentioned before [6,7,8,9]. The magnesium dose recommended by each of the International, American and European Headache Societies and the Neurological Academy [6,7,8,9], is 400–600 mg per day. It seems only slightly more than the dose recommended by The Food and Drug Administration (FDA), which is of around 400 mg for a man and 310 mg for a woman between 19 and 30 years old. However, it should be remembered that the daily intake recommended by the FDA is for a normal diet in healthy people. In addition, there are few RCTs on therapeutic magnesium dosing [60]. In general, oral magnesium supplementation for the treatment of migraine complies with the minimum FDA recommendations, with the exception of 2 RCTs where higher doses are recommended [61]. Of course, the type of the administered substance is also important, and therefore the properties influencing its absorption. Studies on the bioavailability of the different generations of magnesium salts recommend second- and third-generation salts [62,63]. However, the results of the migraine study show the good efficacy of magnesium oxide, and lack of pain relief by second-generation salts [64,65]. The review of Morel et al., covering the issues the bioavailability of magnesium preparations, identified five RCTs that stress the superiority of magnesium citrate or oxide with novel matrices [66], but the superiority of a pharmaceutical form would need further studies, as there are no studies evaluating “head to head” the efficacy of different form of magnesium [66]. On the other hand, we know that the absorption of magnesium from the gastrointestinal tract is the highest in the form of citrate and lactate. It seems justified to recommend those chemical substance as oral preparations [63]. The absorption of the magnesium ion given in the supplements depends not only on its chemical form but also on its solubility in water, dose, method of administration, presence of other absorption enhancers, e.g., the presence of vitamin B6 and potassium ions. Good solubility, stability, and bioavailability make magnesium citrate, compared to other compounds, the most accessible preparation used in magnesium supplementation, both acute and prolonged. The absorption of magnesium when administered as an inorganic compound is anion-dependent and varies in the range of 10–16%. Carbonate, hydroxide, and magnesium oxide are much more difficult to absorb than chloride or magnesium sulfate due to the weak water solubility. The limited biological availability of magnesium oxide, estimated at about 4%, has consistently been shown from numerous studies. However, magnesium oxide is still used due to a significant share in the percentage of magnesium ion in this preparation (60%). Organic magnesium compounds are characterized by greater solubility in water compared to inorganic compounds, which causes much better absorption in the small intestine, reaching the value of 90% [67,68,69]. The highest absorption of magnesium in an organic form was found for its chelate with citrate after oral administration in healthy people.

Finally, the research results of the Khani et al. study, which aimed to assess the efficacy of concurrent therapy with use of sodium valproate and magnesium, and compare it with either sodium valproate or magnesium alone, revealed that magnesium could enhance the antimigraine properties of sodium valproate in combination therapy and reduce the valproate dose for prophylaxis of migraine [69]. It is worth adding that valproates are drugs recommended in class A of prophylactic treatment of both episodic and chronic migraines. Importantly, female patients of childbearing age, as recommended by the European Medicines Agency (EMA), should not take valproate. Thus, it seems that there are grounds for the physician and the young women themselves to prefer magnesium-only therapy, if it brings sufficient clinical benefits, of course. Magnesium preparations, as mentioned earlier, are also used as therapy in menstrual migraine, i.e., a form of migraine in which attacks occur only in the perimenstrual period [70]. Magnesium has also been shown to be additive and to increase the effectiveness of ibuprofen and paracetamol in children and adolescents with migraine attacks. In a study by Gallelli et al. [71], such a positive effect of magnesium was observed in a group of one hundred and sixty children of both sexes aged 5–16 years. The subjects were assigned to four groups: treated with paracetamol or ibuprofen, with or without additional prophylactic use of magnesium. It was found that the better efficacy of pharmacotherapy with the addition of magnesium was independent of the patient’s age (without age-related effects).

## 5. Summary and Conclusions

Magnesium is one of the most important ions involved in cellular/mitochondrial energy changes. It takes part in numerous enzymatic reactions, and also maintains the balance of the membranes of cells, influencing its permeability and reducing the possibility of spontaneous depolarization. It affects the excitability and nerve conduction in the peripheral and also in the central nervous system so plays an important role in migraine course. The results of the research by Slavin et al. [72] suggest that insufficient magnesium consumption is associated with migraine in US adults aged 20–50. This study quantified dietary and total (diet + supplement) magnesium consumption of adults with migraine or severe headaches, and investigated the relationship between magnesium consumption levels and prevalence of migraine or severe headache. Analysis included data from 3626 participants. Authors stated that further prospective investigations are warranted to evaluate the role of dietary magnesium intake in migraine. While maintaining the correct stocks of this ion, proper glutamatergic signaling in the central nervous system is possible. It is necessary for the protective neurons against excitotoxicity and oxidative stress and thus may have an anti-migraine effect [73,74]. Overall, repeating the conclusions of Maier et al. [46] the use of oral magnesium salt represents a well-tolerated and inexpensive addition for the treatment of migraine patients. Supplementation of magnesium may reduce the frequency of attacks, reduce the costs of treatment and reduce serious side effects of treatment. It is important to emphasize that the level of magnesium measured in blood serum does not reflect its concentration in the internal environment of cells (including erythrocytes, platelets, neurons, and myocytes) [74]. Overall, repeating the conclusions of Maier et al. [46] the use of oral magnesium salt represents a well-tolerated and inexpensive addition for the treatment of migraine patients, to reduce the frequency of attacks, to reduce the costs of treatment both adverse. It is important to emphasize that the level of magnesium measured in blood serum does not reflect its content in the internal environment of platelets, erythrocytes, or neurons and myocytes [74]. Cegielska et al. studied [59] a simple indicator of a real magnesium deficiency, which is a positive electrophysiological test for tetany. The results of this research highlight the discrepancy between the level of magnesium measured in the blood serum and an electrophysiological expression, in the form of a positive tetany test result, suggesting an abnormal magnesium balance. The work suggests a possible link to the lack of adequate dietary magnesium supply. It is believed that deficiency of this macromineral occurs in the brains of people with migraine and it may cause the CSD fenmen, which is probably due to the impaired function of blocking the NMDA receptor, AMPA receptor, and glutamatergic stimulation involving the subcortical centers and trigeminal-autonomic system. So, hypomagnesemia may stimulate the neocortex to the hippocampus through the entorhinal cortex and subcortical structures, influencing, apart from triggering the aura, the sensitization of the trigeminal autonomic system (with the effect of enhancing nociceptive sensations) and sensory secretory in the course of an attack [29,30,33,34,35,36]. Therefore, the supply of magnesium in an appropriately absorbable compound and dose can be considered a method of management in patients with migraine, especially those who don’t want or have contraindications to taking drugs recommended in class A/B. This group of patients includes women in the reproductive period, patients with menstrual migraine, or elderly patients with multiple diseases. At the same time, magnesium supplementation can accompany other drugs. We find that our narrative review of studies shows that magnesium has a positive effect in treating migraine at all ages, especially in enhancing the effects of typical antimigraine medications [71].

Table 1 summarizes the studies, the results of which are discussed in the article.

## Figures and Tables

**Figure 1 nutrients-14-01089-f001:**
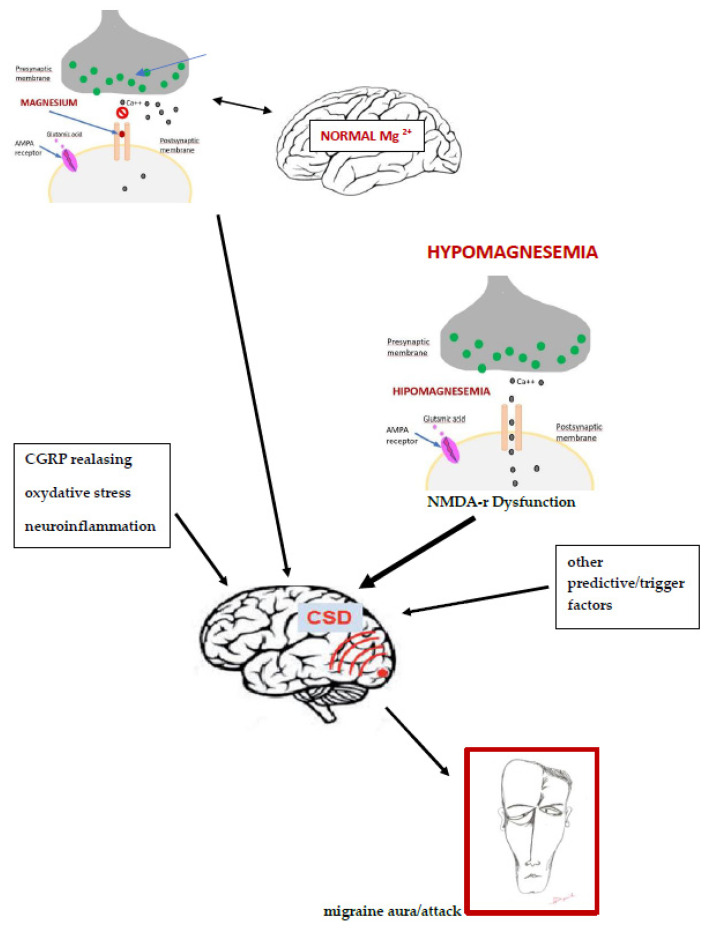
The potential role of magnesium in the pathogenesis of migraine modified according to: [37,38,39].

**Table 1 nutrients-14-01089-t001:** Studies investigating the relationship between magnesium (Mg) levels and migraine.

Reference	Method of Study	Patients No.	Diagnosis	Outcome	Conclusions
Thomas et al., 2000 [53]	case control	29 + 18	migraine + control	significantly lower concentrations of total Mg in erythrocytes and lymphocytes in migraine	Mg in lymphocytes appears to be the most sensitive index of Mg deficiency
Trauninger et.al. 2002 [54]	case control	20 + 20	migraine + healthy	no significant difference between the groups in the baseline serum and urine Mg concentrations, although the latter tended to be lower in the migraine	Mg retention occurs in patients with migraine after oral loading, suggesting a systemic Mg deficiency
Talebi et al., 2011 [55]	case control	140 + 140	migraine + healthy	the average serum Mg level in migraine was significantly lower, no significant difference between the mean level of serum Mg in migraine with aura and without aura	serum Mg in migraine patients was related to the frequency of migraine attacks, supporting the use of Mg in prevention and treatment of migraine
Samaie et al., 2012 [56]	case control	50 + 50	migraine + healthy	no significant differences, but serum total Mg level was notably lower in the migraine	assessing serum Mg level might predict migraine attacks and help to determine optimal dose of administered Mg for achieving appropriate therapeutic outcome
Assarzadegan et al., 2016 [57]	case control	40 + 40	migraine + healthy	significant lower Mg serum levels during the migraine attacks and between the attacks compared with healthy individuals	the serum level of Mg is an independent factor for migraine
Karim et al., 2021 [58]	cross-sectional analytical	70	migraine	serum Mg level lower in severe migraine in comparison to mild to moderate headache	in all migraine groups Mg within normal range
Pfaffenrath et al., 1996 [61]	double-blind placebo-controlled study	150	migraine	with regard to the number of migraine days or migraine attacks there was no benefit with Mg compared to placebo	there were no centre-specific differences, and the final assessments of treatment efficacy by the doctor and patient were largely equivocal
Walker et al., 2003 [62]	randomised double-blind placebo-controlled parallel	46	healthy	supplementation of the organic forms of Mg citrate and amino-acid chelate showed greater absorption than Mg oxide	supplementation with Mg citrate shows superior bioavailability
Karimi et al., 2021 [63]	single-center, randomized, controlled, double-blind, crossover	31 + 32	migraine + control	Mg oxide vs valproate sodium did not show statistically significant difference in the efficacy of both drugs in migraine preventive	Mg oxide can be equally effective in the prevention of migraine attacks as valproate sodium, additionally without significant side effects
Wang et al., 2003 [65]	randomized, double-blind, placebo-controlled, parallel-group	58 + 60 children of ages 3 to 17 years	migraine + control	a statistically significant downward trend in the frequency and severity of migraine pain was found in the group treated with Mg oxide but not in the placebo group	the study is inconclusive
Morel et al., 2021 [66]	systematic review of RCT	81 RCTs on Mg treatment in pain (18 RCTs in migraine)	different types of pain, including migraine	the greatest number of RCTs covering this issue was found in post-operative pain and migraine treated Mg	additional, programmed clinical trials are needed to achieve a sufficient level of scientific evidence to recommend and optimize the use of magnesium in the treatment of pain, mainly chronic pain
Lindberg et al., 1990 [67]	observational	17	healthy	the level of Mg in urine after an oral loading with two Mg salts - higher after Mg citrate than Mg oxide	Mg citrate is more soluble and bioavailable than Mg oxide
Muehlbauer et al., 1991 [68]	observational	24	healthy	Mg-oxide showed significantly lower absorption than Mg-l-aspartate-HCI (granules/tablets)	Mg-l-aspartate-HCI appear to be the first choice for Mg substitution
Khani et al., 2021 [69]	randomized single-center double-blind parallel-group controlled	82 + 70 + 70 (sodium valproate + magnesium with sodium valproate + magnesium)	migraine	significant reduction of migraine in valproate with Mg group	Mg enhance the antimigraine properties of sodium valproate in combination therapy and reduce the required valproate dose for migraine prophylaxis
Gallelli et al., 2014 [71]	single-blinded, balanced-recruitment, parallel-group, single-center	116 children of ages 5–16	migraine	Mg, acetaminophen, ibuprofen decreased pain intensity, but did not modify its frequency; in both acetaminophen and ibuprofen groups, magnesium significantly reduced the pain frequency	Mg increased the efficacy of ibuprofen and acetaminophen with not age-related effects
Slavin et al., 2021 [72]	cross-sectional	3626	migraine or sever headache	Mg consumption associated with lower odds of migraine	evaluate the role of dietary Mg intake on migraine

## Data Availability

Not applicable—the article is a review of previously published research.

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
