# Peer review of "Magnesium as an Important Factor in the Pathogenesis and Treatment of Migraine—From Theory to Practice"

_nutrients, 2022, doi:10.3390/nu14051089_

Round 1

Reviewer 1 Report

Dear Authors,

I have read the manuscript and I send you my comments:

1) Methods: please add the methods used to identify the references

2) Please add this manuscript "  doi: 10.1111/head.12162. Epub 2013 Jun 28." and discuss it "

3) Please add a figure with the mechanism of action of magnesium in headache

4) Please add a table with the characteristics of the patients enrolled in the studies that you descibe

Author Response

Dear Reviewer,

Thank you very much for your thorough evaluation of the work, interest in its methodology and the questions you asked. In response, we explain:

  1. We have made a linguistic correction – corrections are marked in the text.
  2. We have added the methods used to identify the references as follows:

“Our study, as a narrative review, was based on the published articles mainly in recent years. We considered the role of magnesium in the pathogenesis of migraine and therefore its usefulness in the migraine treatment. For this purpose, we took into ac-count the articles that dealt with the above problem and, at the same time,  based on the current studies of on the pathogenetic role of magnesium.”

  1. We have added the proposed manuscript citation (the end of 4th paragraph) and discussed it (the end of discussion/conclusion part) as follows:

“Magnesium has also been shown to be additive and to increase the effectiveness of ibuprofen and paracetamol in children and adolescents with migraine attacks. In a study by Gallelli et al. [72] such a positive effect of magnesium was observed in a group of one hundred and sixty children of both sexes aged 5-16 years. The subjects were assigned to 4 groups: treated with paracetamol or ibuprofen, with or without additional prophylactic use of magnesium. It was found that the better efficacy of pharmacotherapy with the addition of magnesium was independent of the patient's age (without age-related effects).”

and

“We find, that our narrative review of studies shows, that magnesium has a positive effect in treating migraine at all ages, especially in enhancing the effects of typical an-ti-migraine medications [72]. “

  1. We have added the figure with the explanation of the potential role of magnesium in migraine pathogenesis.
  2. We have added the table with description of discussed in our manuscript studies.

Reviewer 2 Report

First of all, English language of the article is way below the acceptable level in grammatical correctness and clarity of meanings. I had to read sentences repeatedly in order to understand what the authors are arguing or claiming, but mostly to fail.

The main argument of the authors seems to be that migraine is caused by electric stimulation in the brain, and it may be prevented by magnesium admistration.  If my understanding is right, figures and/or tables are strongly recommended for the readers to understand the article more efficiently and clearly. 

Author Response

Dear Reviewer,

Thank you very much for your thorough evaluation of the work, interest in its methodology and the questions you asked. In response, we explain:

  1. A native speaker was involved in the editing of the text – corrections are marked in the text –we have changed yellow marked parts.
  2. We have added the figure with the explanation of the potential role of magnesium in migraine pathogenesis.
  3. We have added the table with description of discussed in our manuscript studies.

Round 2

Reviewer 1 Report

None

Reviewer 2 Report

The paper tried to summarize a mjor pathological mechanism of migraine, in which Magnesium plays a great role. By doing so, meaningful therapeutic suggestion was made.

Swift provision of figure(s) and table(s) shows sincerity and passion for the article. 

The review is comprehensive, and its statements and conclusions coherent.